# Does melatonin administration reduce the incidence of postoperative delirium in adults? Systematic review and meta-analysis

Jonathan Barnes,[1] Emma Sewart ![ORCID],[2] Richard A Armstrong ![ORCID],[3] Maria Pufulete,[3] Robert Hinchliffe,[4,5] Ben Gibbison,[1,3] Ronelle Mouton[2,5]

[1]Department of Anaesthesia, Bristol Royal Infirmary, Bristol, UK
[2]Department of Anaesthesia, North Bristol NHS Trust, Bristol, UK
[3]University of Bristol, Bristol, UK
[4]Department of Surgery, North Bristol NHS Trust, Bristol, UK
[5]Department of Population Health Sciences, Bristol Centre for Surgical Research, Bristol, UK

**Correspondence to**
Dr Jonathan Barnes;
Jonathan.barnes4@nhs.net

## ABSTRACT

Postoperative delirium (POD) is common. It is associated with increased morbidity and mortality. Many cases may be preventable and melatonin offers promise as a preventative agent.

**Objective** This systematic review provides an up-to-date synthesis of the evidence on the effect of melatonin in preventing POD.

**Design** A systematic search of randomised controlled trials of melatonin in POD was run across multiple databases (EMBASE, MEDLINE, CINAHL, PsycINFO) and a clinical trials registry (ClinicalTrials.org) (1 January 1990 to 5 April 2022). Studies examining the effects of melatonin on POD incidence in adults are included. Risk of bias was assessed using the Cochrane risk of bias 2 tool.

**Outcome measures** The primary outcome is POD incidence. Secondary outcomes are POD duration and length of hospital stay. Data synthesis was undertaken using a random-effects meta-analysis and presented using forest plots. A summary of methodology and outcome measures in included studies is also presented.

**Results** Eleven studies, with 1244 patients from a range of surgical specialties were included. Seven studies used melatonin, in variable doses, and four used ramelteon. Eight different diagnostic tools were used to diagnose POD. Time points for assessment also varied. Six studies were assessed as low risk of bias and five as some concern. The combined OR of developing POD in the melatonin groups versus control was 0.41 (95% CI 0.21 to 0.80, p=0.01).

**Conclusion** This review found that melatonin may reduce the incidence of POD in adults undergoing surgery. However, included studies displayed inconsistency in their methodology and outcome reporting. Further work to determine the optimum regime for melatonin administration, along with consensus of how best to evaluate results, would be beneficial.

**PROSPERO registration number** CRD42021285019.

## INTRODUCTION

Postoperative delirium (POD) is a common postoperative complication, especially in older patients.[1 2] POD comprises an acute and fluctuating disturbance in attention, awareness and cognition not explained by another neurocognitive disorder.[3] It usually

### STRENGTHS AND LIMITATIONS OF THIS STUDY

⇒ This review provides an up-to-date assessment of the efficacy of melatonin for prevention of postoperative delirium.
⇒ The search was performed across a range of databases and a clinical trials registry to ensure thorough data capture.
⇒ Studies presented in this review include a diverse range of patients, surgical subspecialties and melatonin preparations and administration regimes.
⇒ Results of this review are limited by the heterogeneity of the included studies limits, and lack of consistency of intervention design and outcome measures.
⇒ Language restrictions to English may have led to exclusion of additional studies.

occurs between 1 and 3 days after surgery.[3] Although there is no single universally accepted definition. POD is associated with a number of risk factors, including pain, bleeding, polypharmacy, sedative drugs and major surgery.[4–8] POD carries a significant healthcare and economic burden and is associated with an increased length of hospital stay, increased incidence of ongoing cognitive impairment, increased care needs and increased mortality.[9–11] It is thought that around 30–40% of cases may be preventable[12 13] and strategies to prevent POD have been highlighted as an important priority for healthcare systems.[14]

Melatonin, a hormone produced by the pineal gland, is known to play an important role in circadian rhythm regulation[15] and is an attractive option as an agent for the prevention of POD. It has a low-risk side effect profile and is inexpensive. Melatonin, and melatonin receptor agonists, are used therapeutically in the treatment of sleep-wake disorders[16] and jet lag[17] and may have a role in sedation, analgesia and as an anaesthetic adjunct.[18] Furthermore, melatonin has been

identified as having a potential therapeutic role in several disorders of the mind including schizophrenia, depression and POD.[19–21]

A previous systematic review of six studies (conducted in 2017) examining the use of melatonin and its agonists in the prevention of POD in older adults found some evidence to suggest a beneficial effect.[22] However, these studies were mostly small, heterogeneous in their methodology, and demonstrated conflicting results. Since then, multiple studies investigating melatonin and POD have been published.

The aim of this review is to provide an up-to-date assessment of melatonin as an intervention for the prevention of POD in adult patients. This review will also provide a description of how melatonin has been administered (dose, duration etc.) and how POD has been studied in randomised controlled trials (RCTs).

## METHODS

All results from this work are presented using the Preferred Reporting Items for Systematic Reviews and Meta-Analyses (PRISMA) guidelines and checklist.[23] We prospectively registered the review with PROSPERO and the protocol has been published.[24]

### Search strategy and eligibility criteria

A search was conducted across EMBASE, MEDLINE, CINAHL and PsycINFO for RCTs examining the effects of melatonin, or melatonin receptor agonists, in the prevention of POD. Only studies in adult patients (age 18 years or older) undergoing surgery were included. No restrictions were placed on the type of surgery, the programme of drug administration (including timing, dose and additional treatments used) or on the intervention received by the comparator group. We included studies published in English between 1 January 1990 and 5 April 2022. This window was chosen in line with previous work,[22] as it coincides with the development of the Beers Criteria, which identified medications that may cause or worsen delirium.[25] No limits were placed on the country of study. We also hand searched the reference lists of included studies. The search strategy is detailed in online supplemental appendix 1. The search was repeated using a clinical trials registry (ClinicalTrials.org) to capture ongoing, but unpublished, studies. For any studies identified, and not yet published, the study lead was contacted to request results to be shared.

### Data collection

Citation management and data collection was undertaken using Covidence (Covidence, Melbourne, Australia).[26] Titles and abstracts were independently screened by two reviewers (JB and RM). Full texts were then independently screened by the same two reviewers. All data was extracted in whole by two independent reviewers (JB and RM) using a predefined template. Discrepancies or disagreements at all stages were resolved by discussion between the two main reviewers. If there was still no consensus a third reviewer made the final decision (BG).

### Data items

The key information gathered in the data collection form included:

1. Publication details: authors, country of study and years of study conduct and publication.
2. Participant demographics: sex, age, number, inclusion/exclusion criteria and surgery type.
3. Intervention details: drugs used, dose, timings, details of control interventions.
4. Criteria and scales used for diagnosing and grading POD.

The primary outcome of this review is the incidence of POD in adults undergoing surgery, with incidence of POD reflecting any recorded POD, at any time point. The secondary outcomes are the duration of POD and length of postoperative hospital stay.

### Study risk of bias assessment

We used the Cochrane risk of bias tool, V.2, to assess for risk of bias in the estimated effect of melatonin on POD in each study across five domains[27]: The randomisation process; deviations from the intended interventions (effects of assignment to intervention and effect of adhering to intervention); missing outcome data; measurement of the outcome and selection of the reported results. The trial quality and overall risk of bias were judged as low risk, high risk or some concerns. We inspected a funnel plot and used Egger's linear regression test to detect small study effects.

### Effect measures

A qualitative synthesis and narrative description of the included studies are presented alongside meta-analysis of the primary outcome. All data analysis was undertaken using Stata (StataCorp, Texas, USA).[28] Categorical data are summarised by counts and percentages, and continuous data by means and SD, unless otherwise indicated. Incidence of POD is summarised using ORs (with associated 95% CIs) for individual studies and combined using random effect meta-analysis. Between-study heterogeneity is assessed using the $I^2$ statistic and random-effects estimates and a fixed-effects meta-analysis is also performed as a sensitivity analysis.

### Patient and public involvement

None.

## RESULTS

The search identified 646 studies, with 446 remaining after deduplication. From the clinical trials search 46 studies were identified, 2 were duplicated in the literature search and were excluded. Four hundred and forty-three studies were excluded after abstract review. Of the 47 articles sought for full-text review, 6 were from the clinical trials search. All study leads were contacted. One study

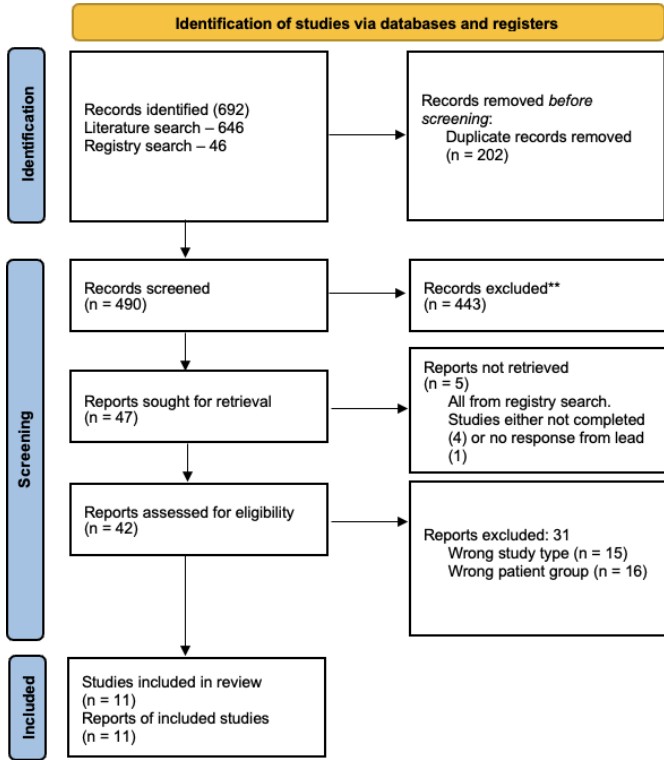

**Identification of studies via databases and registers**

Identification:
- Records identified (692)
  Literature search – 646
  Registry search – 46
  → Records removed *before screening:*
    Duplicate records removed (n = 202)

Screening:
- Records screened (n = 490)
  → Records excluded** (n = 443)
- Reports sought for retrieval (n = 47)
  → Reports not retrieved (n = 5)
    All from registry search. Studies either not completed (4) or no response from lead (1)
- Reports assessed for eligibility (n = 42)
  → Reports excluded: 31
    Wrong study type (n = 15)
    Wrong patient group (n = 16)

Included:
- Studies included in review (n = 11)
  Reports of included studies (n = 11)

**Figure 1** Preferred Reporting Items for Systematic Reviews and Meta-Analyses flow chart for study selection process. N=number of articles. *Consider, if feasible to do so, reporting the number of records identified from each database or register searched (rather than the total number across all databases/registers). **If automation tools were used, indicate how many records were excluded by a human and how many were excluded by automation tools.

was complete but unpublished and the author shared results via personal correspondence.[29] One author did not respond, and four studies were not in a position to share results (either not started recruiting or not completed data collection). After full-text review, 16 articles were excluded due to the patient group (eg, not postoperative patients) and 15 were excluded for being the wrong study type. This process is outlined in a PRISMA flowchart (figure 1). A full list of excluded studies is included in online supplemental appendix 2.

Eleven studies including 1244 participants were eligible for inclusion in this review. Characteristics of these studies and basic participant demographics are outlined in table 1. The studies included patients across a range of surgical specialties, including general, orthopaedic and cardiothoracic surgery. One study included patients from a variety of surgical specialties.[30] Mean patient age was above 50 for all studies and several studies used a minimum age as an exclusion criterion.

Seven studies used melatonin as the study drug, in a variety of doses (from 3 mg to 50 mg/kg and in one study as a long acting formulation), with one using dexmedetomidine alongside melatonin in the intervention arm.[31] Four studies used ramelteon (always 8 mg one time per day). Dosing schedule was highly variable. Three studies included preoperative dosing (two exclusively preoperative, one preoperative and postoperative), in one study medication was given intraoperatively, while the others involved exclusively postoperative dosing. In studies with a priori dosing regimens, dosing programmes ranged from 1 days to 7 days. In two studies patients were included if they had had at least one or three doses, respectively, and in two other studies the duration of treatment was dependent on the length of hospital stay. Eight different measures of delirium were used across the identified studies. One study did not report the assessment tool used,[32] however all other included studies used validated assessment tools. Some studies used multiple tools for diagnosis of delirium and assessing its severity and some tools were used in multiple ways (eg, Confusion Assessment Measure was used a screening tool in one study and a diagnostic tool in others, while the Memorial Delirium Assessment Scale was used a diagnostic tool in one study and a tool to grade severity in another). The time points of assessment were also highly variable (table 1).

Incidence of delirium ranged from 0% to 42% in the intervention groups and 4% to 92% in the control groups. Pooled data showed evidence of a reduced incidence of POD associated with melatonin use (OR 0.41, 95% CI 0.21 to 0.80, 11 studies, 1244 participants, p=0.01). Assuming an overall risk of POD the same as that of the combined study populations (0.23), this gives a number needed to treat of 8.40. Random-effects estimates were used as there is significant between-study heterogeneity ($I^2$=71.97%) (figure 2). However, sensitivity analysis with a fixed-effects meta-analysis gave similar results, also finding a reduced incidence of POD associated with melatonin use (OR 0.63, 95% CI 0.48 to 0.83, p<0.01).

Duration of POD and length of hospital stay were each reported by five studies (table 2). The median durations of delirium reported ranged from 0.7 days[33] to 3 days.[34] Mean duration of hospital stay ranged from 7 days to 17 days. Due to the paucity of reported data, neither synthesis of secondary outcome data nor reporting of complications associated with treatment were performed.

All included studies used some blinding and randomisation process and a validated outcome tool(s). Six studies were considered at low risk of bias, five were rated as 'some concern'. This occurred due to studies lacking either a clearly prespecified analysis plan and/or concerns regarding the blinding process. No studies were considered at high risk of bias (online supplemental appendix 3). No studies were considered at risk of financial bias. Nine studies reported no conflicts of interest. For the other studies, no author reported any support, financial or otherwise, from any source related to the study drug, nor did either study receive any private funding. A funnel plot was prepared and inspected (online supplemental appendix 4) and an Egger's and Harbord test performed. This displayed no clear evidence of small study effects (Egger's p=0.128, Harbord p=0.127).

Certainty of evidence was assessed using the Grading of Recommendations Assessment, Development and

**Table 1** Characteristics of the studies and study participants

| Author (year); country | Year of study conduct | Drug (dose) (all one time per day) I | C | Timing of dosing | Duration of therapy (days) | Study design/ (blinding) | Reason for surgery | Patients N I | C | Male patients N (%) I | C | Age (years) mean (SD) unless otherwise indicated I | C | Assessment of delirium Scale | Time point | Mean (SD) duration of surgery (minutes) I | C |
|---|---|---|---|---|---|---|---|---|---|---|---|---|---|---|---|---|---|
| de Jonghe (2014); The Netherlands | 2008–2012 | Melatonin (3 mg) | Placebo | Postoperative | 5 | RCT (double) | Hip fracture | 186 | 192 | 53 (29) | 62 (32) | 84 (8) | 83 (8) | DSM-IV | Daily for 8 days postoperative (or until discharge) | NR | NR |
| Ford (2020); Australia | NR | Melatonin (3 mg) | Placebo | Postoperative | 7 | RCT (double) | Cardiac | 105 | 105 | 79 (75) | 85 (81) | 69 (8) | 68 (8) | CAM or CAM-ICU (screening); DSM-V (scale) | Daily for 7 days if CAM positive | NR | NR |
| Gupta (2019); Indian | NR | Ramelteon (8 mg) | Placebo | Preoperative | 12 hours | RCT (double) | Mixed (requiring neuraxial anaesthesia) | 50 | 50 | 35 (70) | 33 (66) | 69 (4) | 71 (4) | CAM | Daily for 3 days | NR | NR |
| Hashim (unpublished); Iraq | 2020 | Melatonin (5 mg) | No treatment | Preoperative | Two doses (night before and 90 min preoperative) | RCT (double) | NR | 12 | 12 | NR | NR | >60 | >60 | MDAS | 30, 60 and 90 min postoperative | NR | NR |
| Jaiswal (2019); USA | 2016–2017 | Ramelteon (8 mg) | Placebo | Postoperative | 1–7 | RCT (double) | Elective pulmonary thromboendarterectomy | 59 | 28 | 29 (49) | 29 (50) | 58 (14) | 56 (16) | CAM-ICU and RASS | Two times per day for 9 days (or until ICU discharge) | 526 (480–540)* | 510 (480–540)* |
| Mahrose (2021); Egypt | NR | Melatonin (5 mg) + Dexmedetomidine | Dexmedetomidine (bolus+infusion) | Preoperative/ postoperative | 4 | RCT (NR) | Elective coronary artery bypass graft | 55 | 55 | 42 (76) | 41 (75) | 67 (7) | 66 (6) | CAM or CAM-ICU | Two times per day for 5 days (or until discharge) | NR | NR |
| Nickkholgh (2011); Germany | 2007–2009 | Melatonin (50 mg/kg) | Placebo | Intraoperative | 1 | RCT (double) | Liver resection | 25 | 23 | 17 (68) | 11 (48) | 59 (10) | 56 (11) | NR | Up to 7 days (frequency NR) | 202 (80) | 212 (79) |
| Oh (2020); USA | 2017–2019 | Ramelteon (8 mg) | Placebo | Postoperative | 3 | RCT (quadruple) | Elective primary or revision hip or knee replacement | 41 | 39 | 14 (42) | 14 (37) | 74 (6) | 75 (5) | DSM-V | Daily for 3 days | 138 (72) | 162 (84) |
| Sultan (2010); Egypt | NR | Melatonin (5 mg) | No treatment | Postoperative | 2 | RCT (double) | Hip arthroplasty | 53 | 49 | 24 (45) | 22 (45) | 70 (7) | 72 (6) | AMT | Daily for 3 days | 127 (45) | 120 (37) |
| Yamaguchi† (2014); Japan | NR | Ramelteon (8 mg) | Placebo | Postoperative | 4 | RCT (double) | Total knee arthroplasty | 22 | 23 | NR | NR | ≥70 | ≥70 | iCDSC | Three times per day for 4 days | NR | NR |
| Zadeh (2021); Iran | 2018–2019 | Melatonin (3 mg) prolonged release | Placebo | | 4 | RCT (double) | Coronary artery bypass graft | 30 | 30 | 20 (67) | 22 (73) | 60 (10) | 63 (8) | CAM-ICU and MDAS | Two times per day for 48 hours post extubation | NR | NR |

*Median (IQR).
†Abstract only.
‡Median (range).
AMT, Abbreviated Mental Test; C, control group; CAM, Confusion Assessment Method; CAM-ICU, Confusion Assessment Method for the intensive care unit; DSM, diagnostic and statistical manual of mental disorders; I, intervention group; iCDSC, intensive care delirium screening checklist score; kg, kilograms; MDAS, Memorial Delirium Assessment Scale; mg, milligrams; NR, not reported; RASS, Richmond Agitation Sedation Scale; RCT, randomised control trial.

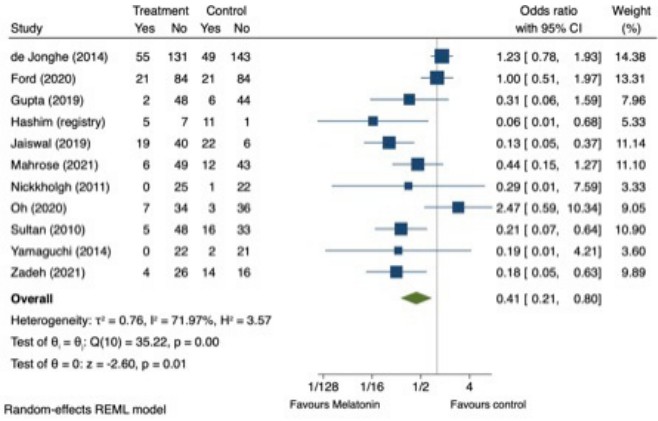

**Figure 2** Forest plot displaying ORs and 95% CIs of effect of melatonin or melatonergic agent on incidence of postoperative delirium per study. Overall OR generated using a random-effects meta-analysis.

Evaluation approach. Results are outlined in table 3. Overall, evidence included in this review supports melatonin as effective prophylaxis for delirium with a moderate level of certainty. Recommendation was rated down due to heterogeneity/inconsistency, which could not be explained as subgroup analysis was unable to be undertaken.

## DISCUSSION

The main findings from this systematic review are that melatonin and melatonin receptor agonists are associated with a reduced incidence of POD in adult patients undergoing surgery, while huge variation exists in the intervention design as well as the outcome measures employed.

The development of POD is multifactorial. Risk factors include patient factors (eg, pre-existing cognitive impairment, atherosclerosis), intraoperative factors (eg, use of benzodiazepines, high stress surgery) and postoperative factors (eg, disrupted sleep-wake cycling, pain).[35][36] The aetiology is thought to include a direct neuroinflammatory process, accompanied by disruption in various neurotransmitter pathways.[36][37] Melatonin may act to reduce POD both by ensuring, and restoring, normal sleep-wake cycling along with exerting a direct anti-inflammatory action,[38] involving free radical scavenging, immune modulation and direct neuroprotection.[39]

Despite the rationale for melatonin preventing POD, a recent RCT found no impact of melatonin on incidence of POD in adult intensive care unit (ICU) patients.[40] However, this study assessed only critically unwell patients who started taking melatonin treatment after they had been admitted to ICU. This therefore does not rule out potential benefits of melatonin in different patient groups and in different treatment regimens. The results from our review concur with the findings from two previous, smaller reviews on postoperative patient populations.[22][41] These reviews both suggest melatonin as an effective prevention for POD, but are limited by small numbers of heterogenous studies. Since both of these

| Table 2 | Study outcomes | | | | | |
|---|---|---|---|---|---|---|
| **Author (year); country** | **Delirium incidence N (%)** | | **Delirium duration (days) median (range)** | | **Hospital stay (days) mean (SD) unless otherwise indicated** | |
| | I | C | I | C | I | C |
| de Jonghe (2014); The Netherlands | 55 (30) | 49 (25.5) | 2 (1–3) | 2 (1–3) | 11 (6–14.5)* | 11 (8–17)* |
| Ford (2020); Australia | 21 (21) | 21 (20) | 3 (2–4)† | 2 (1–3)† | 8 (6–10)† | 7 (6–8)† |
| Gupta (2019); Indian | 2 (4) | 6 (12) | NR | NR | NR | NR |
| Hashim (unpublished); Iraq | 5 (42) | 11 (92) | NR | NR | NR | NR |
| Jaiswal (2019); USA | 19 (32) | 22 (38) | 1.0 (0.6–1.5)† | 0.7 (0.4–1.2)† | 12 (10–16)† | 12 (10–14)† |
| Mahrose (2021); Egypt | 6 (11) | 12 (27) | 1.0 (±0.3) | 2.0 (±0.6) | 12 (±5) | 14 (±6) |
| Nickkholgh (2011); Germany | 0 | 1 (4) | NR | NR | 13.5 (±1.5) | 17 (±2) |
| Oh (2020); USA | 7 (21)‡ | 3 (8)‡ | 1.3 (±0.5) | 1.2 (±0.5) | NR | NR |
| Sultan (2010); Egypt | 5 (9) | 16 (33) | NR | NR | NR | NR |
| Yamaguchi (2014); Japan | 0 | 2 (9) | NR | NR | NR | NR |
| Zadeh (2021); Iran | 4 (13)‡ | 14 (47)‡ | NR | NR | NR | NR |

*Median (range).
†Median (IQR).
‡Delirium incidence reported at more than one time point. The highest reported rate is included here.
C, control group; I, intervention group; NR, not reported.

**Table 3** GRADE assessment of evidence certainty, melatonin compared with placebo for prevention of postoperative delirium

| Melatonin compared with placebo for prevention of postoperative delirium | | | | | |
|---|---|---|---|---|---|
| Patient or population: Adult patients undergoing surgery.<br>Setting: Hospital.<br>Intervention: Melatonin or melatonin receptor agonist.<br>Comparison: Placebo. | | | | | |
| Outcomes | Anticipated absolute effects | | Relative effect (95% CI) | № of participants (studies) | Certainty of the evidence (GRADE) | Comments |
| | Risk with placebo | Risk with melatonin | | | | |
| Delirium | Study population | | OR 0.41 (0.21 to 0.80) | 1244 (11 RCTs) | ⊕⊕⊕ MODERATE | Low risk of bias, precise, no evidence of publication bias. Rated down for inconsistency. Delirium incidence was recorded at multiple time points, this refers to 'any delirium' during the study window. |
| | 157/606 | 124/638 | | | | |

GRADE Working Group grades of evidence.
High certainty: We are very confident that the true effect lies close to that of the estimate of the effect.
Moderate certainty: We are moderately confident in the effect estimate: The true effect is likely to be close to the estimate of the effect, but there is a possibility that it is substantially different.
Low certainty: Our confidence in the effect estimate is limited: The true effect may be substantially different from the estimate of the effect.
Very low certainty: We have very little confidence in the effect estimate: The true effect is likely to be substantially different from the estimate of effect.

*The risk in the intervention group (and its 95% CI) is based on the assumed risk in the comparison group and the relative effect of the intervention (and its 95% CI).
GRADE, Grading of Recommendations Assessment, Development and Evaluation; RCTs, randomised controlled trials; RR, risk ratio.

reviews, there has been further interest in the effects of melatonin on POD, allowing us to include multiple additional RCTs in this review. The reduction of POD seen with melatonin in our work is also in keeping with the results from a mixed patient review, which concluded that 'melatonin/ramelteon are associated with reduction in delirium… however this effect seems confined to surgical and ICU patients'.[42]

Multiple databases were searched, using a standardised search, along with hand searching of references and a registry search, ensuring good capture of includable studies. No included studies were at high risk of bias and those included captured a wide variety of patient groups and surgical specialties. Alongside qualitative analysis of the available evidence this study also adds quantitative evaluation of synthesised data on the effect of melatonin on preventing POD.

Our review is limited by the heterogeneity of the published trials of melatonin for the prevention of POD. The study interventions were highly variable, including populations, drugs used, dosing and administration regimens. The outcome measures used varied widely across studies, and there was little consistency in when assessments were undertaken, or how frequently. Given delirium has a fluctuant nature, this may have contributed to under-reporting of delirium in several trials. The lack of consistency of either intervention or outcome measure also makes it challenging to conclude what an optimal melatonin dosing schedule would look like, and evaluation of this would be a valuable area for further study. This heterogeneity, along with the small number of included studies, also made it impractical to perform a sensitivity analysis of results, further limiting the strength of our results. Most studies excluded patients with pre-existing cognitive impairment and those on known psychoactive medications. These patients represent some of the highest risk of POD and would therefore arguably be some of those most likely to benefit from POD reduction strategies. Furthermore, the interventions used in the included studies may not reflect the likely use of melatonin in practice, limiting the external validity of these results. It is likely that future POD prevention strategies will involve multifaceted care bundles, rather than single interventions (eg, non-pharmaceutical sleep hygiene strategies, proactive pain management, avoidance of other exacerbating medications along with any pharmaceutical agent(s)). A final limitation was that we were unable to undertake subgroup analyses due to patient numbers. This would be an area for future study, particularly in the cardiac surgical population, who, due to the nature of the surgery, anaesthesia and recovery, are a group at particularly high risk of POD.[43 44]

In conclusion, melatonin has both biological plausibility and clinical evidence to support it as a treatment to prevent POD. However, the optimum protocol for melatonin use is yet to be determined. A consensus on how

best to deliver the intervention and evaluate results is needed before further, adequately-powered clinical evaluation can be performed to properly analyse melatonin as a preventative intervention for delirium. Clinical applicability of findings from future work may be enhanced by focusing on those at highest risk of POD.

**Correction notice** This article has been corrected since it first published. Funder grant number has been added and the open access licence type has been changed to CC BY. 14th September 2023.

**Acknowledgements** We thank Professor Jonathan Sterne, professor of medical statistics and epidemiology, for his assistance with assessment of potential reporting bias and Sarah Rudd, Knowledge Specialist, for her assistance with the literature search.

**Contributors** JB and ES are joint first authors and RM and BG are joint final authors. All authors have contributed fully to the concept and design of the review for which this protocol has been written. JB, ES, BG and RM wrote and reviewed the manuscript before submission. RAA supervised the meta-analysis. MP, RAA and RH reviewed and edited the manuscript before submission. RM is the guarantor of the review.

**Funding** This work was supported by the NIHR Biomedical Research Centre at University Hospitals Bristol and Weston NHS Foundation Trust and the University of Bristol and British Heart Foundation (CH/1992027/7163). RAA and ES are NIHR Academic Clinical Fellows

**Disclaimer** The views expressed in this publication are those of the author(s) and not necessarily those of the NHS, the National Institute for Health Research, the Department of Health and Social Care.

**Competing interests** None declared.

**Patient and public involvement** Patients and/or the public were not involved in the design, or conduct, or reporting, or dissemination plans of this research.

**Patient consent for publication** Not applicable.

**Provenance and peer review** Not commissioned; externally peer reviewed.

**Data availability statement** No data are available. As a systematic review and meta-analysis, all data used is from other articles. No original research data was generated.

**ORCID iDs**
Emma Sewart http://orcid.org/0000-0002-1214-0356
Richard A Armstrong http://orcid.org/0000-0001-9479-0143

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
