## [Reviewer comments · BMJ Open]

ARTICLE DETAILS

TITLE (PROVISIONAL)	Does melatonin administration reduce the incidence of postoperative delirium in adults? Systematic Review and Meta-analysis
AUTHORS	Barnes, Jonathan; Sewart, Emma; Armstrong, Richard; Pufulete, Maria; Hinchliffe, Robert; Gibbison, Ben; Mouton, Ronelle

VERSION 1 – REVIEW

REVIEWER	Nair, Balakrishnan Newcastle University, John Hunter Campus
REVIEW RETURNED	05-Dec-2022

GENERAL COMMENTS	Postoperative delirium may be the most common complication; however, it is underdiagnosed and undertreated. Melatonin is cheap, free of any serious side -effects and may be beneficial. So this paper is very timely; it is well-written and thought out.
--

REVIEWER	Sigaut, Stéphanie Hôpital Beaujon, Anesthesiology and Intensive Care
REVIEW RETURNED	13-Dec-2022

GENERAL COMMENTS	The authors propose a meta-analysis evaluating melatonin for postoperative delirium (POD) prevention. The last MA on this subject is five years old, and studies have been published since. Moreover, POD is a major public health issue, and melatonin is a serious pharmacological contender for its prevention, with a solid physiopathological rationale. Thus, this MA is of interest. It is well-written and easy to read, with up-to-date bibliography. Nevertheless, before publication, it could be improved by adding the following: - A formal analysis of how potential biases interfere with the results of the meta-analysis- A list of excluded studies- Detail of potential differences in search strategies between different databases- the risk of financial bias for the selected studies- a paragraph in the Discussion section about POD physiopathology (essentially neuroinflammation induced by surgical inflammatory stress) and the link with the rationale for melatonin effect (immunomodulatory effects).- a paragraph discussing the particularity POD after cardiac surgery (physiopathology, incidence...) and how results in this population may differ or not from other surgical populations due to these particularities Moreover, the funnel plot interpretation should be rewritten :
--

	 - Several studies are outside the triangle or touching the margin, especially at the top, where the most powerful studies are figured, suggesting a risk of bias - The plot is clearly asymmetrical and suggest missing studies at the bottom right-hand side of the plot
--	--

VERSION 1 – AUTHOR RESPONSE

Reviewer: 1

Prof. Balakrishnan Nair, Newcastle University

Comments to the Author:

Postoperative delirium may be the most common complication; however, it is underdiagnosed and undertreated. Melatonin is cheap, free of any serious side -effects and may be beneficial. So this paper is very timely; it is well-written and thought out.

Thank you for your feedback and for taking the time to review this paper.

Reviewer: 2

Dr. Stéphanie Sigaut, Hôpital Beaujon

Comments to the Author:

The authors propose a meta-analysis evaluating melatonin for postoperative delirium (POD) prevention. The last MA on this subject is five years old, and studies have been published since. Moreover, POD is a major public health issue, and melatonin is a serious pharmacological contender for its prevention, with a solid physiopathological rationale. Thus, this MA is of interest. It is well-written and easy to read, with up-to-date bibliography.

Thank you for your feedback.

Nevertheless, before publication, it could be improved by adding the following:

- A formal analysis of how potential biases interfere with the results of the meta-analysis

As a group, after long discussion, the authors agreed that a sensitivity analysis would not be appropriate in this meta-analysis. This was due to the small number of studies, alongside the dichotomous nature of the outcome studied (i.e. delirium yes/no) coupled with the highly heterogenous nature of the included studies – in terms of intervention, outcome measure and patient group. Having considered several options no clear methodological clusters existed, hence we felt it would not be appropriate to perform a sensitivity analysis. We have expanded our discussion to give clarification of this point.

- A list of excluded studies

We have added in, as an appendix, a list of studies excluded after full text review, along with the primary reason for exclusion.

- Detail of potential differences in search strategies between different databases

The search strategies were essentially identical across databases and we have included these as an appendix. This has also been noted in the discussion.

- the risk of financial bias for the selected studies

None of the studies were felt to display a risk of financial bias. Nine reported no conflicts of interest.

For the other two, the study was not funded by any private company, and the disclosures made by individual authors did not involve any company or entity related to the study drug (intervention).

We have added this into our results.

- a paragraph in the Discussion section about POD pathophysiology (essentially neuroinflammation induced by surgical inflammatory stress) and the link with the rationale for melatonin effect (immunomodulatory effects).

Thank you for this suggestion. Our discussion has been expanded to include a fuller description of the mechanisms underlying POD, and how melatonin may exert its potential therapeutic effect.

- a paragraph discussing the particularity POD after cardiac surgery (pathophysiology, incidence...) and how results in this population may differ or not from other surgical populations due to these particularities

We have expanded our discussion to specifically discuss POD after cardiac surgery.

Moreover, the funnel plot interpretation should be rewritten :

- Several studies are outside the triangle or touching the margin, especially at the top, where the most powerful studies are figured, suggesting a risk of bias

- The plot is clearly asymmetrical and suggest missing studies at the bottom right-hand side of the plot
The funnel plot results were discussed at length between the authors, with expert input from Professor Jonathan Sterne, professor of medical statistics. We have reviewed the plot again and performed both an Egger's and Harbord test on the data, which showed little evidence of small study effects (Egger's $p=0.128$, Harbord $p=0.127$). We have expanded our results section to clarify this explanation of our funnel plot interpretation.

Reviewer: 1

Competing interests of Reviewer: Nil

Reviewer: 2

Competing interests of Reviewer: None